# Capsaicin as a Dietary Additive for Dairy Cows: A Meta-Analysis on Performance, Milk Composition, Digestibility, Rumen Fermentation, and Serum Metabolites

**DOI:** 10.3390/ani14071075

**Published:** 2024-04-02

**Authors:** José Felipe Orzuna-Orzuna, Juan Eduardo Godina-Rodríguez, Jonathan Raúl Garay-Martínez, Alejandro Lara-Bueno

**Affiliations:** 1Departamento de Zootecnia, Universidad Autónoma Chapingo, Chapingo C.P. 56230, Mexico; jforzuna@gmail.com; 2Campo Experimental Uruapan, Instituto Nacional de Investigaciones Forestales, Agrícolas y Pecuarias, Av. Latinoamérica 1001, Uruapan C.P. 60150, Michoacán, Mexico; godina.juan@inifap.gob.mx; 3Campo Experimental Las Huastecas, Instituto Nacional de Investigaciones Forestales, Agrícolas y Pecuarias, Altamira C.P. 89610, Tamaulipas, Mexico; garay.jonathan@inifap.gob.mx

**Keywords:** bioactive compounds, alkaloids, capsicum oleoresin, nutrient digestibility

## Abstract

**Simple Summary:**

The demand for cow’s milk has increased in recent years due to the rapid growth in the world’s human population. The objective of this study was to evaluate the effects of dietary supplementation with the natural alkaloid capsaicin on milk production and composition, nutrient digestibility, rumen fermentation, and serum metabolites of dairy cows using meta-analytical statistical procedures. The results indicate that dietary supplementation with capsaicin increases dry matter intake, milk production, feed efficiency, milk fat content, nutrient digestibility, and ruminal production of total volatile fatty acids. Likewise, blood metabolites indicate better energy balance in cows supplemented with capsaicin. In conclusion, the inclusion of capsaicin in diets for dairy cows can help improve milk production and composition, nutrient digestibility, ruminal fermentation, and blood metabolites.

**Abstract:**

This study aimed to evaluate the effects of dietary supplementation with capsaicin (CAP) on productive performance, milk composition, nutrient digestibility, ruminal fermentation, and serum metabolites of dairy cows using a meta-analytical approach. The database included 13 studies, from which the response variables of interest were obtained. Data were analyzed using a random effects model, and results were expressed as weighted mean differences between treatments supplemented with and without CAP. Dietary supplementation with CAP increased (*p* < 0.05) dry matter intake, milk yield, feed efficiency, milk fat yield, and milk fat content. However, CAP supplementation did not affect (*p* > 0.05) milk protein and lactose yield, milk urea nitrogen, or milk somatic cell count. Greater (*p* < 0.05) apparent digestibility of dry matter and crude protein was observed in response to the dietary inclusion of CAP. Likewise, supplementation with CAP increased (*p* < 0.05) the rumen concentration of total volatile fatty acids. In contrast, CAP supplementation did not affect (*p* > 0.05) ruminal pH or the ruminal concentration of ammonia nitrogen, acetate, propionate, and butyrate. In blood serum, CAP supplementation increased (*p* < 0.05) the glucose concentration and decreased (*p* < 0.05) the concentration of non-esterified fatty acids. However, CAP supplementation did not affect (*p* > 0.05) the serum concentration of urea and beta-hydroxybutyrate. In conclusion, capsaicin can be used as a dietary additive to improve the productive performance, milk composition, and nutrient digestibility in dairy cows and, at the same time, improve the ruminal concentration of total volatile fatty acids and serum levels of glucose and non-esterified fatty acids.

## 1. Introduction

Cow’s milk is the most consumed worldwide [1] and is considered a complete human food due to its wide content of essential nutrients, such as proteins, fats, vitamins, and minerals [2]. According to some authors [3,4], in recent years, the demand for cow’s milk has increased due to the growth of the world population. Therefore, to satisfy this growing demand for milk, it is necessary to increase the productive performance of dairy cows. Firkins and Mitchell [5] mention that, through the dietary inclusion of food additives, such as yeasts, seaweed, and plant extracts, it is possible to improve digestibility, ruminal fermentation, and productive efficiency in dairy cows without negatively affecting the milk composition. Among the plant extracts currently available is capsaicin, which has recently shown promising effects in improving the productive performance of dairy cattle [6,7].

According to Xiang et al. [8], CAP (trans-8-methyl-N-vanillyl-6-nonenamide) is an alkaloid found mainly in the fruits of *Capsicum* spp. and represents approximately 69% of the total capsaicinoids present in these fruits. CAP improves lipid metabolism and has various properties such as antioxidant, anti-inflammatory, antimicrobial, thermoregulatory, and anticancer [9,10]. To date, the use of CAP in human and animal nutrition has been limited because its strong pungency can cause gastrointestinal upset [8], even if it is also dose-dependent [11]. However, in recent studies, CAP has been successfully used to improve the growth performance, digestibility, and health of pigs [12,13] and broilers [14,15]. On the other hand, in beef cattle that are fed diets high in concentrate, some feed additives containing CAP stimulate feed consumption without affecting ruminal fermentation [16,17]. Likewise, the dietary inclusion of CAP decreases the ruminal concentration of ammonia nitrogen without changing the proportion of short-chain fatty acids in beef cattle fed high-forage diets. Furthermore, supplementation with high doses (4 mg/kg DM) of CAP improves the productive performance, milk composition, and serum metabolites in lactating buffaloes [18].

In particular, in dairy cows, few studies have been conducted to evaluate the effects of CAP as a dietary additive [6,19,20]. Furthermore, the results observed in some of these studies are inconsistent, which prevents reliable conclusions from being obtained. For example, some authors [7,19] did not detect significant changes in the productive performance and milk composition of dairy cows supplemented with low CAP doses (0.04 and 0.10 mg/kg DM). Likewise, other authors observed negative effects on milk yield [21] and nutrient digestibility [22] of dairy cows supplemented with CAP. In contrast, recent studies [6,23,24] reported a positive impact of the consumption of low doses (0.05 and 0.28 mg/kg DM) and high doses (>20 mg/kg DM) of CAP on the productive performance and milk composition of dairy cows. According to Adaszek et al. [9], most of the biological effects of CAP consumption in humans and animals depend on the dose and duration of the experimental period.

Some recent narrative reviews [9,25] show that alkaloids, including CAP, have the potential as feed additives in domestic animals to improve productive performance, metabolism, and general health status. However, none of these reviews focused on dairy cows, and neither evaluated the effects of CAP on milk composition, nutrient digestibility, or ruminal fermentation. Furthermore, some authors [26] mention that traditional reviews commonly lead to biased conclusions because they do not quantitatively evaluate the studies included in them. In contrast, according to several authors [27,28,29], meta-analysis is an effective method to quantitatively analyze studies on the same topic with contradictory findings and obtain reliable conclusions. To our knowledge, the effects of dietary CAP supplementation in dairy cows have not been evaluated by meta-analysis. The hypothesis of the present study establishes that dietary supplementation with CAP will positively impact the productive performance, milk composition, nutrient digestibility, rumen fermentation, and serum metabolites of dairy cows. Consequently, the objective of this study was to evaluate the effects of dietary capsaicin supplementation on productive performance, milk composition, nutrient digestibility, rumen fermentation, and serum metabolites of dairy cows using a meta-analytical approach.

## 2. Materials and Methods

### 2.1. Literature Search

To formulate the research question, the Population, Intervention, Comparison, and Outcomes (PICO) strategy was used [30]. Briefly, the population was dairy cows, the intervention was dietary supplementation with CAP, the comparison was between diets with and without CAP, and the result was each of the values obtained in productive performance, milk composition, nutrient digestibility, fermentation rumen, and serum metabolites. Peer-reviewed scientific articles that evaluated the effects of dietary supplementation with CAP in dairy cows following PRISMA guidelines [31] were identified, selected, chosen, and included in the final database, as shown in Figure 1. For this, systematic searches of scientific documents were carried out in the Web of Science, Scopus, PubMed, and ScienceDirect databases. In each database, the following keywords were used: capsaicin, dairy cows, milk production, milk composition, nutrient digestibility, ruminal fermentation, and serum metabolites. Searches were limited to studies published in English between January 2011 and December 2023.

### 2.2. Inclusion and Exclusion Criteria

Initially, the searches returned 101 scientific articles, which were reduced to 80 after removing duplicate articles. First, articles that had one or more of the following exclusion criteria were removed from the database: (1) studies that did not use CAP or used CAP combined with artificial sweeteners, essential oils, or another dietary additive; (2) studies that did not use dairy cows; and (3) theses, literature reviews, conference proceedings, or books. Subsequently, to be included in the final database, the remaining articles had to meet similar inclusion requirements as those used by other authors [28,32,33]: (1) full-text scientific articles published in English in peer-reviewed journals; (2) studies that used dairy cows as experimental animals; (3) studies that measured and reported data on productive performance, milk composition, nutrient digestibility, ruminal fermentation, or serum metabolites; (4) studies that evaluated the impact of CAP supplementation versus a control treatment using the same basal diet; (5) studies that reported the amount of CAP (mg/kg DM) included in the diet or provided the information necessary to estimate it; and (6) studies that included data on treatment means, number of experimental units (n), and standard error of means (SEM) of control (diets without CAP supplementation) and experimental (diets supplemented with CAP) treatments.

### 2.3. Data Extraction

The final database consisted of 13 scientific articles, listed in Table 1. Each of these articles was organized in an Excel spreadsheet using the first author and year of publication. Subsequently, the following information was extracted from each document: (1) breed of dairy cows; (2) days in milk that the cows had; (3) CAP dose (mg/kg DM); (4) duration of CAP supplementation period (days); and (5) amount of forage (g/kg DM) included in the experimental diets. The final database only included response variables reported in at least three scientific articles, as recommended by other authors [27,28,34]. The included variables were grouped as follows: (1) dry matter intake (DMI), milk yield (MY), milk yield corrected to 4% fat content (4FCMY), feed efficiency (FE), milk fat yield (MFY), milk protein yield (MPY), milk lactose yield (MLY), milk fat content (MFC), milk protein content (MPC), milk lactose content (MLC), total solids (TS) in milk, milk urea nitrogen (MUN), and milk somatic cell count (SCC); (2) apparent dry matter digestibility (ADMD), apparent organic matter digestibility (AOMD), apparent crude protein digestibility (ACPD), apparent neutral detergent fiber digestibility (ANDFD), apparent acid detergent fiber digestibility (AADFD), ammonia nitrogen (NH_3_-N), total volatile fatty acids (TVFA), acetate, propionate, butyrate, and valerate; (3) glucose, urea, beta-hydroxybutyrate (BHB), and non-esterified fatty acids (NEFA). Treatment means, SEM, and n were extracted for each response variable.

### 2.4. Calculations and Statistical Analysis

The meta [40] and metafor [29] packages of the R statistical software (version 4.1.2) were used in the statistical analyses. In all response variables, the effects of CAP supplementation were evaluated using weighted mean differences (WMD). WMDs were chosen because, according to Takeshima et al. [41], they improve the interpretability of the results. WMDs were calculated as the means of the experimental treatments (diets added with CAP) minus the means of the control treatments (without the addition of CAP in the diets). Studies were weighted by the inverse of the variance using the methods of Der-Simonian and Laird [42] for random effects models.

### 2.5. Heterogeneity and Publication Bias

The presence of heterogeneity was evaluated with the chi-square (Q) test using a significance level of ≤0.05 [43]. In addition, the I^2^ statistic was used to detect and quantify treatment heterogeneity [34]. According to Borenstein et al. [44], the I^2^ values can be interpreted as follows: (1) I^2^ ≤ 25% is low heterogeneity; (2) I^2^ of 26% to 75% is moderate heterogeneity; and (3) I^2^ > 75% is high heterogeneity. The possible presence of publication bias was evaluated through Egger’s regression asymmetry test [45] and Begg’s adjusted rank correlation [46]. For these two tests, publication bias was declared when *p* ≤ 0.05.

## 3. Results

### 3.1. Milk Yield and Composition

Table 2 shows that DMI, MY, 4FCMY, FE, MFY, and MFC increased (*p* < 0.05) in response to CAP supplementation. However, CAP supplementation did not affect (*p >* 0.05) MPY, MLY, MPC, MLC, TS, MUN, and SCC.

### 3.2. Nutrient Digestibility and Ruminal Fermentation

Table 3 shows that CAP supplementation increased (*p* < 0.05) ADMD, ACPD, AADFD, and TVFA. However, CAP supplementation did not affect (*p* > 0.05) AOMD, ANDFD, ruminal pH, and ruminal concentration of NH_3_-N, acetate, propionate, butyrate, and valerate.

### 3.3. Serum Metabolites

Dietary inclusion of CAP increased (*p* < 0.05) serum glucose concentration (Table 4). However, CAP supplementation did not affect (*p* > 0.05) serum levels of urea and BHB. In contrast, serum NEFA concentration decreased (*p* < 0.05) in response to dietary CAP supplementation.

### 3.4. Heterogeneity and Publication Bias

Table 2 and Table 4 show that there was no heterogeneity (Q) (*p* ≤ 0.05) in any of the response variables evaluated. In contrast, Table 3 shows significant Q (*p* ≤ 0.05) in ADMD, AOMD, ADNFD, and AADFD. However, several authors [28,33,34] recommend not applying meta-regression analysis on response variables reported in less than ten studies because, in these conditions, the power of the test is low. Therefore, meta-regression was not used in the present study. On the other hand, Table 2, Table 3 and Table 4 show that Egger’s regression asymmetry test and Begg’s adjusted rank correlation were not significant (*p* > 0.05) in any of the response variables tested, suggesting no publication bias.

## 4. Discussion

### 4.1. Milk Yield and Composition

In the present meta-analysis, dietary inclusion of CAP increased DMI. Similar effects were previously reported in beef cattle supplemented with CAP and fed diets high in forage [15] and concentrate [16]. In the current study, the higher DMI observed in cows supplemented with CAP could be related to the increases detected in ADMD, ACPD, and AADFD, as suggested by Vitorazzi et al. [39]. Although the mechanism through which CAP regulates DMI in ruminants is unclear, in other mammals, short-term CAP use stimulates DMI through changes in the sensory systems of the vagus nerve [47].

Greater MY, FE, 4FCMY, MFY, and MFC were observed in response to CAP supplementation. Similarly, Abulaiti et al. [18] detected greater MY and MFC in lactating buffaloes supplemented with increasing levels (2, 4, and 6 mg/kg DM for 30 and 45 days) of CAP. Likewise, Cunha et al. [48] reported higher MY, FE, and MFC in lactating sheep supplemented with CAP (0.5 and 1.0 mg/kg DM for 18 days). In the present meta-analysis, CAP supplementation improved DMI, ADMD, ACPD, and TVFA, which suggests greater ingestion, digestion, and metabolic availability of nitrogenous compounds and energy that could be used for milk production and partially explain the positive effects observed in MY and FE. Likewise, in Holstein calves [49] and non-ruminants [11,14], supplementation with CAP increases the serum concentration of immunoglobulins A (IgA), growth hormone (GH), glutathione peroxidase (GPX), and superoxide dismutase (SD). Similar effects of CAP consumption in dairy cows could be related to the observed increases in MY and FE since a recent study [50] reported that serum levels of IgG, GH, SOD, and CAT are positively correlated (r between 0.56 and 0.64) with MY and FE in dairy cows.

On the other hand, the greater MFY and MFC observed in the present meta-analysis are positive since, according to De Oca-Flores et al. [51], MFY and MFC are positively correlated (r between 0.33 and 0.39) with cheese yield. The increases detected in 4FCMY and MFY could be explained by the greater MFC observed in cows supplemented with CAP since, according to Wongpom et al. [52], MFC has a positive correlation (r between 0.40 and 0.70) with 4FCMY and MFY. The exact mechanism by which CAP increases MFC in dairy cows has not yet been studied. However, in other mammals, CAP stimulates the mobilization of lipids from adipose tissue and leads to higher serum concentrations of free fatty acids [48,53]. A similar effect of CAP consumption in dairy cows could increase blood levels of fatty acids, which could subsequently be used by the mammary gland to synthesize milk fat [54]. On the other hand, in the present study, the lack of changes observed in MUN suggests that the dietary inclusion of CAP does not affect the utilization of rumen-degradable protein in the diet [55]. Likewise, the lack of changes detected in SCC indicates that CAP consumption does not affect udder health or milk quality in dairy cows [56].

### 4.2. Nutrient Digestibility and Ruminal Fermentation

In the current study, CAP supplementation increased ADMD, ACPD, and AADFD. In non-ruminants, several recent studies [12,14] show that dietary supplementation with CAP increases the production and secretion of digestive enzymes (α-amylase, trypsin, chymotrypsin, and gastric and pancreatic lipase) and the length of intestinal villi. Therefore, similar effects of CAP consumption in dairy cows could explain the higher ADMD and ACPD detected in the present study. On the other hand, CAP consumption increases the ruminal relative abundance of *Fibrobacter* bacteria in dairy cows between 11.9 and 20.5% [36]. Likewise, Oh et al. [37] detected between 7.3 and 13.2% greater presence of *Ruminococcus* bacteria in dairy cows supplemented with CAP. A greater ruminal abundance of *Fibrobacter* and *Ruminococcus* bacteria could lead to greater AADFD since these microorganisms can hydrolyze the cellulose in the feed [57].

In ruminant nutrition studies, evaluating rumen fermentation parameters is important because they serve as indicators of nutrient digestion and metabolism by rumen microorganisms [58]. In the current study, CAP supplementation increased the ruminal concentration of TVFA without altering the ruminal pH or the ruminal concentration of NH_3_-N, acetate, propionate, butyrate, and valerate. Similarly, other studies reported that dietary inclusion of CAP increased the rumen concentration of TVFA without significantly changing other short-chain fatty acids in the ruminal fluid of beef cattle [58] and calves [49]. The highest concentration of TVFA detected in the present meta-analysis is positive since, according to Seymour [59], TVFA represents between 60 and 70% of the metabolizable energy in ruminants, and the ruminal concentration of TVFA is positively correlated (r = 0.52) with MY in dairy cows [59]. On the other hand, in the present meta-analysis, the lack of significant changes in ruminal pH suggests that CAP consumption does not affect the internal homeostasis of the ruminal environment [28]. Likewise, the lack of changes detected in the ruminal concentration of NH_3_-N indicates that the dietary inclusion of CAP does not affect the utilization of ammonia by rumen microorganisms or the balance between the release and absorption of ammonia in the rumen [58,60].

### 4.3. Serum Metabolites

Serum metabolites can be used as reliable markers to evaluate protein and energy metabolism in dairy cows [61]. In the current study, CAP supplementation increased serum glucose, indicating a positive effect on energy metabolism [61,62]. Similarly, recent studies detected higher serum glucose in buffaloes [18] and lactating sheep [48] supplemented with CAP. Some authors reported that CAP consumption decreases serum insulin concentration in dairy cows [35,37]. This CAP effect partially explains the lower serum glucose observed because, according to Azarbayejani and Mohammadsadegh [63], serum glucose and insulin levels are negatively correlated (r = 0.52) in Holstein cows.

On the other hand, CAP supplementation did not affect serum urea and BHB levels; however, the concentration of NEFA in blood serum decreased. The lack of changes in serum urea suggests that CAP does not affect protein anabolism in dairy cows [49]. This effect was expected since, in the current study, CAP did not modify the ruminal concentration of NH_3_-N, and this parameter has a linear relationship (r = 0.55) with serum urea in ruminants [60]. Likewise, the lack of changes in ruminal NH_3_-N concentration explains the lack of changes observed in MUN because these two parameters have a positive relationship [55]. Likewise, Dong et al. [50] indicated that serum BHB levels are mainly used to diagnose ketosis in dairy cows. Therefore, in the current study, the lack of changes detected in BHB suggests that CAP does not modify the incidence of ketosis. Furthermore, the lower serum NEFA concentration detected in the present meta-analysis is positive because serum NEFA levels have a high negative correlation (r = −0.72) with energy balance in dairy cows [64]. The reduction in serum NEFA could be related to the observed increase in serum glucose since, according to Zamuner et al. [62], serum NEFA and glucose levels are negatively correlated (r = −0.54) in dairy ruminants.

## 5. Conclusions

Dietary supplementation with capsaicin stimulates dry matter intake and increases milk yield, feed efficiency, milk fat content, and milk fat yield. Likewise, capsaicin supplementation improves the apparent digestibility of dry matter, crude protein, and acid detergent fiber and the rumen concentration of total volatile fatty acids. The results of serum metabolites indicate that capsaicin can improve energy balance in dairy cows through an increase in serum glucose concentration and a reduction in serum levels of non-esterified fatty acids.

## Figures and Tables

**Figure 1 animals-14-01075-f001:**
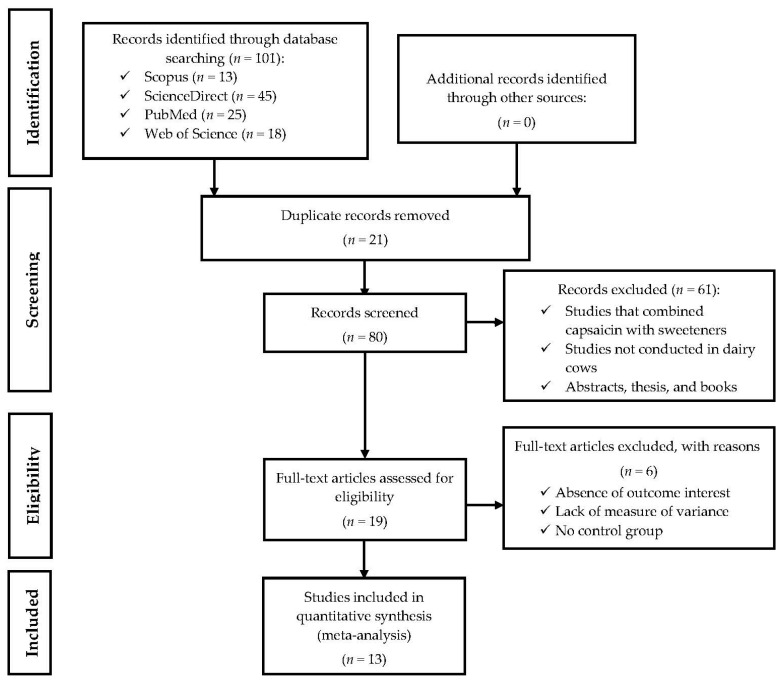
A PRISMA flow diagram detailing the literature search strategy and study selection for the meta-analysis.

**Table 1 animals-14-01075-t001:** Description of the studies included in the meta-analysis database.

Reference	Breed	Days in Milk	Days of Experiment	Dose (mg/kg DM)	Forage (g/kg DM)
Abulaiti et al. [24]	Holstein	34	45	20, 40, 60	620
An et al. [35]	Holstein	150	30	2, 4, 8	407
Foskolos et al. [22]	Holstein	130	31	1.41	437
Grigoletto et al. [23]	Holstein	182	21	0.28	480
Oh et al. [21]	Holstein	175	23	5.63	622
Oh et al. [36]	Holstein	50	25	0.11, 22, 45	510
Oh et al. [37]	Holstein	100	28	0.03, 0.06	555
Oh et al. [19]	Holstein	120	49	0.10	590
Oh et al. [7]	Holstein	0	80	0.04	540
Peretti et al. [38]	Holstein	147	21	0.16, 0.16	480
Tager and Krause [20]	Holstein	43	21	0.65	417
Takiya et al. [6]	Holstein	0	84	0.05	480
Vitorazzi et al. [39]	Holstein	150	63	0.15, 0.29	480

**Table 2 animals-14-01075-t002:** Milk yield and composition of dairy cows supplemented with capsaicin.

Item	N (NC)				Heterogeneity		Egger Test ^1^	Begg Test ^2^
		Control Means (SD)	WMD (95% CI)	*p*-Value	*p*-Value	I^2^ (%)	*p*-Value	*p*-Value
DMI, kg/d	12 (19)	25.16 (2.98)	0.295 (0.020; 0.571)	0.036	0.288	13.60	0.490	0.435
MY, kg/d	13 (21)	34.08 (8.75)	1.181 (0.565; 1.797)	<0.001	0.934	0.00	0.121	0.497
4FCMY, kg/d	8 (14)	34.56 (4.90)	1.414 (0.274; 2.554)	0.015	0.224	36.49	0.864	0.697
FE, MY/DMI	10 (15)	1.570 (0.063)	0.054 (0.019; 0.089)	0.002	0.155	27.37	0.553	0.589
MFY, kg/d	12 (19)	1.39 (0.24)	0.080 (0.044; 0.115)	<0.001	0.826	0.00	0.466	0.339
MPY, kg/d	12 (19)	1.41 (0.63)	0.020 (−0.015; 0.055)	0.254	0.111	47.76	0.413	0.304
MLY, kg/d	11 (16)	1.87 (0.39)	0.029 (−0.017; 0.076)	0.217	0.128	34.33	0.669	0.060
MFC, g/100 g	13 (21)	3.65 (0.65)	0.122 (0.044; 0.199)	0.002	0.167	23.38	0.414	0.269
MPC, g/100 g	13 (21)	3.14 (0.26)	0.002 (−0.017; 0.021)	0.827	0.997	0.00	0.985	0.117
MLC, g/100 g	12 (18)	4.75 (0.16)	−0.010 (−0.040; 0.020)	0.513	0.119	36.93	0.809	0.620
TS, g/100 g	5 (10)	11.51 (0.98)	0.018 (−0.147; 0.183)	0.831	0.902	0.00	0.219	0.222
MUN, mg/dL	8 (15)	14.05 (4.04)	−0.155 (−0.593; 0.284)	0.490	0.560	0.00	0.471	0.472
SCC, ×10^3^ cell/mL	7 (13)	2.80 (1.38)	−0.136 (−0.369; 0.097)	0.253	0.938	0.00	0.879	0.831

N: number of studies; NC: number of comparisons between the capsaicin treatment and control treatment; SD: standard deviation; WMD: weighted mean differences between the control and treatments with capsaicin; CI: confidence interval of WMD; *p*-value to χ2 (Q) test of heterogeneity; I^2^: proportion of total variation of size effect estimates that is due to heterogeneity; ^1^: Egger’s regression asymmetry test; ^2^: Begg’s adjusted rank correlation; DMI: dry matter intake; MY: milk yield; 4FCMY: 0.4 (kg of milk) + 15.0 (kg of fat); FE: feed efficiency; MFY: milk fat yield; MPY: milk protein yield; MLY: milk lactose yield; MFC: milk fat content; MPC: milk protein content; MLC: milk lactose content; TS: total solids; MUN: milk urea nitrogen; SCC: somatic cell count.

**Table 3 animals-14-01075-t003:** Nutrient digestibility and ruminal fermentation of dairy cows supplemented with capsaicin.

Item	N (NC)				Heterogeneity		Egger Test ^1^	Begg Test ^2^
		Control Means (SD)	WMD (95% CI)	*p*-Value	*p*-Value	I^2^ (%)	*p*-Value	*p*-Value
Nutrient digestibility								
ADMD, g/100 g	8 (13)	65.98 (5.98)	0.820 (0.061; 1.578)	0.034	<0.001	97.88	0.917	0.235
AOMD, g/100 g	8 (13)	67.62 (6.37)	0.693 (−0.145; 1.530)	0.105	<0.001	98.44	0.928	0.107
ACPD, g/100 g	8 (13)	65.76 (6.70)	2.174 (1.959; 2.389)	<0.001	0.116	48.71	0.948	0.470
ANDFD, g/100 g	8 (12)	47.62 (12.16)	0.452 (−1.298; 2.202)	0.613	0.002	63.09	0.822	0.062
AADFD, g/100 g	4 (8)	43.58 (11.89)	1.341 (−0.078; 2.761)	0.044	<0.001	98.26	0.974	0.913
Ruminal fermentation								
Ruminal pH	4 (6)	5.93 (0.22)	−0.046 (−0.140; 0.049)	0.343	0.916	0.00	0.931	0.270
NH_3_-N, mg/dL	4 (6)	17.76 (5.01)	0.059 (−1.521; 1.638)	0.942	0.736	0.00	0.397	0.687
TVFA, mM	4 (6)	131.02 (13.59)	0.998 (0.610; 1.386)	<0.001	0.971	0.00	0.744	0.999
Acetate, mol/100 mol	4 (6)	56.55 (6.32)	−0.235 (−1.254; 0.785)	0.652	0.951	0.00	0.569	0.421
Propionate, mol/100 mol	4 (6)	26.08 (5.58)	0.198 (−0.960; 1.357)	0.737	0.976	0.00	0.574	0.688
Butyrate, mol/100 mol	4 (6)	12.60 (0.35)	0.067 (−0.839; 0.974)	0.884	0.992	0.00	0.378	0.321
Valerate, mol/100 mol	4 (6)	2.47 (0.66)	−0.239 (−1.081; 0.603)	0.578	0.974	0.00	0.320	0.462

N: number of studies; NC: number of comparisons between the capsaicin treatment and control treatment; SD: standard deviation; WMD: weighted mean differences between the control and treatments with capsaicin; CI: confidence interval of WMD; *p*-value to χ2 (Q) test of heterogeneity; I^2^: proportion of total variation of size effect estimates that is due to heterogeneity; ^1^: Egger’s regression asymmetry test; ^2^: Begg’s adjusted rank correlation; ADMD: apparent dry matter digestibility; AOMD: apparent organic matter digestibility; ACPD: apparent crude protein digestibility; ANDFD: apparent neutral detergent fiber digestibility; AADFD: apparent acid detergent fiber digestibility; NH_3_-N: ammonia nitrogen; TVFA: total volatile fatty acids.

**Table 4 animals-14-01075-t004:** Serum metabolites of dairy cows supplemented with capsaicin.

Item	N (NC)				Heterogeneity		Egger Test ^1^	Begg Test ^2^
		Control Means (SD)	WMD (95% CI)	*p*-Value	*p*-Value	I^2^ (%)	*p*-Value	*p*-Value
Glucose, mg/dL	8 (15)	58.78 (22.83)	4.053 (0.479; 7.626)	0.026	0.127	46.67	0.432	0.730
Urea, mg/dL	5 (9)	23.10 (11.15)	0.137 (−0.806; 1.079)	0.776	0.185	42.35	0.190	0.332
BHB, μmol/L	5 (9)	681.20 (261.60)	29.772 (−1.775; 61.318)	0.064	0.998	0.00	0.069	0.201
NEFA, μmol/L	4 (9)	278.50 (122.50)	−14.614 (−28.264; −0.965)	0.036	0.163	45.90	0.397	0.153

N: number of studies; NC: number of comparisons between the capsaicin treatment and control treatment; SD: standard deviation; WMD: weighted mean differences between the control and treatments with capsaicin; CI: confidence interval of WMD; *p*-value to χ2 (Q) test of heterogeneity; I^2^: proportion of total variation of size effect estimates that is due to heterogeneity; ^1^: Egger’s regression asymmetry test; ^2^: Begg’s adjusted rank correlation; BHB: beta-hydroxybutyrate; NEFA: non-esterified fatty acids.

## Data Availability

The datasets used and analyzed during the current study are available from the corresponding author upon reasonable request.

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
