# Peer review of "Capsaicin as a Dietary Additive for Dairy Cows: A Meta-Analysis on Performance, Milk Composition, Digestibility, Rumen Fermentation, and Serum Metabolites"

_animals, 2024, doi:10.3390/ani14071075_

Round 1
Reviewer 1 Report
Comments and Suggestions for Authors
Manuscript number: Animals-2941315
Title: Capsaicin as a dietary additive for dairy cows: A meta-analysis on performance, milk composition, digestibility, rumen fermentation, and serum metabolites.
General comment:
Although the style and grammar of this manuscript are praiseworthy, it nonetheless exhibits a considerable number of ambiguities. Particularly in names, aims, achievements, and conclusions that lack an obvious correlation. The PRISMA technique does not provide an explanation for the overall number of studies. The study intended for publication in the "Animals Journal" should exhibit the attributes of lucidity and captivation.
Abstract:
- Refer to the comment embedded within the text.
- Keyword: bioactive compounds; alkaloids; capsicum oleoresin? Should be similar to title!
Introduction:
· L 67-68: Provide a concise overview of the impact of capsaicin (CAP) specifically on dairy cattle, to avoid the misconception that many animals are being discussed in this paragraph.
· L 76 dan L 80: If you write a "positive impact" on L80 then it would be better to write a "negative impact" on L76 so that it correlates the same.
· L77 dan L 81: The good effect shown at L81 may be attributed to the administration of large doses of CAP, but the negative effect observed at L77 cannot be accounted for by the use of high doses of CAP.
· L 90: Add the relevance of meta-analysis study to assess the usage of CAP in dairy cows because of inconsistent outcomes from earlier studies.
Material and method:
· L113-115: Are several keywords used typed in one search?
· L118: During the "included journal (n=13)" stage of the PRISMA technique, how many studies are included in the 13 pieces of literature? The PRISMA technique is a systematic approach used to conduct and report on systematic reviews and meta-analyses. It provides a clear and structured framework for researchers to follow, ensuring transparency and accuracy in the review process.
· Please supply the descriptive statistics for all the studies (literature) that were used.
· Provide an explanation of the constraints of a research. Can the findings of a meta-analysis be considered very reliable if the value of NC is below 10?
Result and Discussion
· L247-250: What is the correlation between buffalo and sheep in this comparative study?
· Where can I get discussions pertaining to performance? The title and purpose are mentioned, however they are not elaborated upon individually in the findings and discussion section. Kindly provide a more explicit explanation.
· For the purpose of conversation, it is important to focus on the method and significance of the topic, rather than making comparisons with other species.
· The explanation is ambiguous and lacks clarity.
Conclusion:
The conclusions lack an examination of the specified objectives. Kindly verify whether performance parameters are present; if they are, they should be distinguished in the results. The clarity of the improvement results remains limited due to inadequate explanations of the findings.
References:
Check format style and Journal Abbreviation!
Considering this, I believe that the present format must be unambiguous to meet the criteria for publishing in the Animals Journal.
Comments on the Quality of English Language
Moderate editing of English language required
Author Response
Response to Reviewer #1:
We would like to thank the reviewer for the careful and thorough reading of this manuscript and for the thoughtful comments and constructive suggestions, which help to improve the quality of this manuscript. Our response follows (the reviewer's comments are in italics).
Comments and Suggestions for Authors
Title: Capsaicin as a dietary additive for dairy cows: A meta-analysis on performance, milk composition, digestibility, rumen fermentation, and serum metabolites.
General comment:
Although the style and grammar of this manuscript are praiseworthy, it nonetheless exhibits a considerable number of ambiguities. Particularly in names, aims, achievements, and conclusions that lack an obvious correlation. The PRISMA technique does not provide an explanation for the overall number of studies. The study intended for publication in the "Animals Journal" should exhibit the attributes of lucidity and captivation.
Response: Dear reviewer, thank you for your comments. Please find our responses to your specific comments below.
Abstract
Comment 1. - Refer to the comment embedded within the text.
Response: There is no comment embedded in the text and neither did any added.
Comment 2. - Keyword: bioactive compounds; alkaloids; capsicum oleoresin? Should be similar to title!
Response: Capsicum oleoresin is one of the main sources of capsaicin. Capsaicin is an alkaloid, and alkaloids are bioactive compounds. By adding these keywords related to capsaicin, we ensure a wide reach for our manuscript in academic search engines. If we add keywords similar to those in the title as the reviewer suggests, the reach of our manuscript in academic search engines will be reduced. Therefore, the reviewer's recommendation is not scientifically correct or properly justified.
Introduction
Comment 3. · L 67-68: Provide a concise overview of the impact of capsaicin (CAP) specifically on dairy cattle, to avoid the misconception that many animals are being discussed in this paragraph.
Response: Dear reviewer, please review lines 73-82 again. A broad overview of the impact of capsaicin specifically on dairy cows has been described in these lines since the first version of our manuscript. Also, in response to your suggestion, lines 68-70 have been removed.
Comment 4. · L 76 dan L 80: If you write a "positive impact" on L80 then it would be better to write a "negative impact" on L76 so that it correlates the same.
Response: We disagree with the reviewer's comment. Lines 76-80 begin by mentioning studies that observed no effects of capsaicin in dairy cows. Subsequently, scientific evidence is shown that found negative effects of capsaicin. Finally, these negative effects are contrasted by showing evidence of the positive effects of capsaicin in dairy cows. That is, the lines have an order and include a lack of effects, negative effects, and positive effects. Therefore, we do not understand the scientific basis of the reviewer's comment.
Comment 5. · L77 dan L 81: The good effect shown at L81 may be attributed to the administration of large doses of CAP, but the negative effect observed at L77 cannot be accounted for by the use of high doses of CAP.
Response: We do not understand the objective of this comment. We never mentioned in the introduction that the negative effect observed in L77 could be explained by the use of high doses of CAP.
Comment 6. · L 90: Add the relevance of meta-analysis study to assess the usage of CAP in dairy cows because of inconsistent outcomes from earlier studies.
Response: We do not understand why the reviewer makes this comment. From the first version of the manuscript, we added the relevance of the meta-analysis study to evaluate inconsistent results from previous studies and obtain reliable results. This justification had been added on lines 91-93. Please check again and carefully.
Material and methods
Comment 7. · L113-115: Are several keywords used typed in one search?
Response: Of course, several keywords can be used in a search.
Comment 8. · L118: During the "included journal (n=13)" stage of the PRISMA technique, how many studies are included in the 13 pieces of literature? The PRISMA technique is a systematic approach used to conduct and report on systematic reviews and meta-analyses. It provides a clear and structured framework for researchers to follow, ensuring transparency and accuracy in the review process.
Response: 13 studies were included, as shown in Figure 1. This figure mentions that the studies included in the quantitative synthesis were 13. Please review carefully.
Comment 9. · Please supply the descriptive statistics for all the studies (literature) that were used.
Response: There is no scientific basis to support this request. Descriptive statistics of the included studies are not useful and are never reported in these types of studies. Below, we mention a brief list of meta-analyses published in MDPI (and other prestigious journals) journals in which it can be observed that descriptive statistics of the studies are never added to the manuscripts:
Mendoza-Martínez, G.D.; Orzuna-Orzuna, J.F.; Roque-Jiménez, J.A.; Gloria-Trujillo, A.; Martínez-García, J.A.; Sánchez-López, N.; Hernández-García, P.A.; Lee-Rangel, H.A. A Polyherbal Mixture with Nutraceutical Properties for Ruminants: A Meta-Analysis and Review of BioCholine Powder. Animals 2024, 14, 667. https://doi.org/10.3390/ani14050667
Plata-Pérez, G.; Angeles-Hernandez, J.C.; Morales-Almaráz, E.; Del Razo-Rodríguez, O.E.; López-González, F.; Peláez-Acero, A.; Campos-Montiel, R.G.; Vargas-Bello-Pérez, E.; Vieyra-Alberto, R. Oilseed Supplementation Improves Milk Composition and Fatty Acid Profile of Cow Milk: A Meta-Analysis and Meta-Regression. Animals 2022, 12, 1642. https://doi.org/10.3390/ani12131642
Wei, C.; He, T.; Wan, X.; Liu, S.; Dong, Y.; Qu, Y. Meta-Analysis of Rumen-Protected Methionine in Milk Production and Composition of Dairy Cows. Animals 2022, 12, 1505. https://doi.org/10.3390/ani12121505
He, T.; Wei, C.; Lin, X.; Wang, B.; Yin, G. Meta-Analysis of the Effects of Organic Chromium Supplementation on the Growth Performance and Carcass Quality of Weaned and Growing-Finishing Pigs. Animals 2023, 13, 2014. https://doi.org/10.3390/ani13122014
Comment 10. · Provide an explanation of the constraints of a research. Can the findings of a meta-analysis be considered very reliable if the value of NC is below 10?
Response: It is more important to consider how many studies reported the response variable instead of considering NC since we may have a large number of NC but reported in a low number of studies or the opposite case. Some authors of basic literature on meta-analysis indicate that results can be considered reliable when they are reported in at least three different studies (Littell et al., 2008). Last but not least, in lines 145-146 of our manuscript, it was mentioned that only response variables reported in at least three studies were included in the database.
Littell, J.H.; Corcoran, J.; Pillai, V. Systematic Reviews and Meta-Analysis, 1st ed.; Oxford University Press: Oxford, UK, 2008; pp. 111–132.
Results and discussion
Comment 11. · L247-250: What is the correlation between buffalo and sheep in this comparative study?
Response: We never mentioned that these species were correlated. However, due to the limited number of studies on capsaicin in dairy cows, we evaluated the consistency of our results by comparing them with those of sheep and buffalo. This comparison is valid since the sheep and buffalo studies were also conducted using lactating females (as indicated in the manuscript), as in our meta-analysis.
Comment 12. · Where can I get discussions pertaining to performance? The title and purpose are mentioned, however they are not elaborated upon individually in the findings and discussion section. Kindly provide a more explicit explanation.
Response: We do not understand why the reviewer mentions that performance findings are not detailed. These findings were shown from the first version of the manuscript in lines 182-185 and Table 2. Likewise, the performance data were discussed in the discussion section in lines 239-259. Please check again.
Comment 13. For the purpose of conversation, it is important to focus on the method and significance of the topic, rather than making comparisons with other species.
Response: The comparison of results with other similar species is useful, especially when there is limited information on the species studied (as in the case of our manuscript). Furthermore, the instructions for authors of the journal Animals MDPI indicate that the discussion can be done considering previous studies (https://www.mdpi.com/journal/animals/instructions).
Comment 14. · The explanation is ambiguous and lacks clarity.
Response: Dear reviewer, the information available on the effects of capsaicin in dairy cows is very limited as indicated by the low number (n = 13) of studies available to date. Therefore, it is not possible to expand the explanation further. In fact, capsaicin is even less studied in other ruminant species than in dairy cows.
Conclusion
Comment 15. The conclusions lack an examination of the specified objectives. Kindly verify whether performance parameters are present; if they are, they should be distinguished in the results. The clarity of the improvement results remains limited due to inadequate explanations of the findings.
Response: The conclusion has been rewritten in a more specific form according to the objectives and results shown in the results section. Lines 329-334. The conclusion now says: “Dietary supplementation with capsaicin stimulates dry matter intake and increases milk yield, feed efficiency, milk fat content, and milk fat yield. Likewise, capsaicin supplementation improves the apparent digestibility of dry matter, crude protein, acid detergent fiber, and the rumen concentration of total volatile fatty acids. The results of serum metabolites indicate that capsaicin can improve energy balance in dairy cows through an increase in serum glucose concentration and a reduction in serum levels of non-esterified fatty acids”.
References
Comment 16. Check format style and Journal Abbreviation!
Response: The original version of the manuscript was prepared using the instructions for authors available on the Animals Journal website. Furthermore, the reviewer made no specific comments indicating any formatting errors.
Reviewer 2 Report
Comments and Suggestions for Authors
It is an interesting work on the effect of Capsaicin on productive parameters and rumen fermentation, although it has a conclusion that I think needs to be reconsidered.

Author Response
Response to Reviewer #2:
We would like to thank the reviewer for careful and thorough reading of this manuscript and for the thoughtful comments and constructive suggestions, which help to improve the quality of this manuscript. Our response follows (the reviewer's comments are in italics).
Comments and Suggestions for Authors
It is an interesting work on the effect of Capsaicin on productive parameters and rumen fermentation, although it has a conclusion that I think needs to be reconsidered.
Response: The conclusion has been rewritten in a more specific form according to the results shown in the results section. Lines 329-334. The conclusion now says: “Dietary supplementation with capsaicin stimulates dry matter intake and increases milk yield, feed efficiency, milk fat content, and milk fat yield. Likewise, capsaicin supplementation improves the apparent digestibility of dry matter, crude protein, acid detergent fiber, and the rumen concentration of total volatile fatty acids. The results of serum metabolites indicate that capsaicin can improve energy balance in dairy cows through an increase in serum glucose concentration and a reduction in serum levels of non-esterified fatty acids”.
Comment 2: Change standard deviation by standard error.
Response: Regarding the suggestion to change standard deviations for standard errors, we do not agree with this suggestion. The traditional and most scientifically correct way to show the means of control treatments is using standard deviations. Below is a short list of previous meta-analyses published in MDPI journals that reported standard deviations instead of standard errors:
Mendoza-Martínez, G.D.; Orzuna-Orzuna, J.F.; Roque-Jiménez, J.A.; Gloria-Trujillo, A.; Martínez-García, J.A.; Sánchez-López, N.; Hernández-García, P.A.; Lee-Rangel, H.A. A Polyherbal Mixture with Nutraceutical Properties for Ruminants: A Meta-Analysis and Review of BioCholine Powder. Animals 2024, 14, 667. https://doi.org/10.3390/ani14050667
Torres, R.d.N.S.; da Silva, D.A.V.; Chardulo, L.A.L.; Baldassini, W.A.; de Almeida, R.A.T.; Almeida, M.T.C.; Curi, R.A.; Pereira, G.L.; Schoonmaker, J.P.; Machado Neto, O.R. The Impact of Liver Abscesses on Performance and Carcass Traits in Beef Cattle: A Meta-Analysis Study. Ruminants 2024, 4, 79-89. https://doi.org/10.3390/ruminants4010005
Orzuna-Orzuna, J.F.; Lara-Bueno, A. Essential Oils as a Dietary Additive for Laying Hens: Performance, Egg Quality, Antioxidant Status, and Intestinal Morphology: A Meta-Analysis. Agriculture 2023, 13, 1294. https://doi.org/10.3390/agriculture13071294
Reviewer 3 Report
Comments and Suggestions for Authors
Please see the attachment

Comments on the Quality of English Language
Author Response
Response to Reviewer #3:
We would like to thank the reviewer for careful and thorough reading of this manuscript and for the thoughtful comments and constructive suggestions, which help to improve the quality of this manuscript. Our response follows (the reviewer's comments are in italics and blue).
Comments and Suggestions for Authors
The authors aimed to evaluate the effects of dietary capsaicin supplementation on milk yield and chemical composition as well as on nutrient utilization and serum metabolites of dairy cows using a meta-analytical approach.
The topic is very important mainly in the intensive breeding system, where it is very difficult to meet the high nutritive requirements with the usuals diets thus generating several metabolic disorders.
The meta-analytycal approach allows to quantitative evaluate the results obtained by different researches thus better describe a phenomenon.
The authors used very well the methodologies and very well depicted and discussed their results adding useful scientific informations. The tables are well comprehensible and evincive.
The conclusions are highly congruent with the results.
Response: Thank you very much for your valuable and careful review of our manuscript.
Comment 1. I suggest to publish the paper just after the following modification and/or additions:
line 61: after the number 8 into the brackets, please add the sentence: “even if it is also dose dependent (Morittu et al., 2020)”.
Please add the following:
Morittu et al. 2020. Potential beneficial and/or adverse effects of Capsicum annuum L. (cv. Fiesta) at two stage of ripening in CD-1 mice. Natural Product Research, 34 (11), 1647-1651
Response: This change has been made, lines 61-62. Also, the suggested reference has been added in the list of references (Lines 374-376).
Comment 2. line 316: in order to facilitate the reader to connect the different results, please write a sentence which correlate those of lines 270-272 and 299-302.
Response: The following sentence correlating lines 270-271 and 299-302 has been added as suggested (see lines 318-320): “Likewise, the lack of changes in ruminal NH3-N concentration explains the lack of changes observed in MUN because these two parameters have a positive relationship [55]”.
Round 2
Reviewer 1 Report
Comments and Suggestions for Authors
Thank you for considering my comment and improving the manuscript. There is a minor concern remaining and requires revision before acceptance.
L41-42 and other line: "improve ruminal fermentation and 41 serum metabolites" Please specify what kind of rumen fermentation or serum metabolites. Not all of it was improved, right?
Author Response
Response to Reviewer #1:
We would like to thank the reviewer for the careful and thorough reading of this manuscript and for the thoughtful comments and constructive suggestions, which helped to improve the quality of this manuscript. Our response follows (the reviewer's comments are in italics).
Comments and Suggestions for Authors
Thank you for considering my comment and improving the manuscript. There is a minor concern remaining and requires revision before acceptance.
Comment 1. L41-42 and other line: "improve ruminal fermentation and 41 serum metabolites" Please specify what kind of rumen fermentation or serum metabolites. Not all of it was improved, right?.
Response: The following sentence has been added on lines 41-42 to specify how ruminal fermentation and serum metabolites were improved: “improve the ruminal concentration of total volatile fatty acids and serum levels of glucose and non-esterified fatty acids”.